# Advances in the Mechanisms of Plant Tolerance to Manganese Toxicity

**DOI:** 10.3390/ijms20205096

**Published:** 2019-10-14

**Authors:** Jifu Li, Yidan Jia, Rongshu Dong, Rui Huang, Pandao Liu, Xinyong Li, Zhiyong Wang, Guodao Liu, Zhijian Chen

**Affiliations:** 1Institute of Tropical Crop Genetic Resources, Chinese Academy of Tropical Agriculture Sciences, Haikou 571101, China; li_jifu@163.com (J.L.); YidanJia@163.com (Y.J.); dongrongshu@126.com (R.D.); bluesing@126.com (R.H.); liupandao@foxmail.com (P.L.); lixy051985@163.com (X.L.); liuguodao2008@163.com (G.L.); 2Institute of Tropical Agriculture and Forestry, Hainan University, Haikou 571101, China; wangzhiyong7989@163.com; 3Key Laboratory of Crop Gene Resources and Germplasm Enhancement in Southern China, Ministry of Agriculture, Hainan 571101, China

**Keywords:** manganese toxicity, Mn detoxification, tolerance mechanism, gene function, subcellular compartment

## Abstract

Manganese (Mn) is an essential element for plant growth due to its participation in a series of physiological and metabolic processes. Mn is also considered a heavy metal that causes phytotoxicity when present in excess, disrupting photosynthesis and enzyme activity in plants. Thus, Mn toxicity is a major constraint limiting plant growth and production, especially in acid soils. To cope with Mn toxicity, plants have evolved a wide range of adaptive strategies to improve their growth under this stress. Mn tolerance mechanisms include activation of the antioxidant system, regulation of Mn uptake and homeostasis, and compartmentalization of Mn into subcellular compartments (e.g., vacuoles, endoplasmic reticulum, Golgi apparatus, and cell walls). In this regard, numerous genes are involved in specific pathways controlling Mn detoxification. Here, we summarize the recent advances in the mechanisms of Mn toxicity tolerance in plants and highlight the roles of genes responsible for Mn uptake, translocation, and distribution, contributing to Mn detoxification. We hope this review will provide a comprehensive understanding of the adaptive strategies of plants to Mn toxicity through gene regulation, which will aid in breeding crop varieties with Mn tolerance via genetic improvement approaches, enhancing the yield and quality of crops.

## 1. Introduction

Manganese (Mn) is the second most prevalent trace element in the Earth’s crust after iron (Fe), and is widely distributed in soils, sediments, and other biological materials [1]. In soils, Mn is present in a wide range of oxidation states, including Mn(II), Mn(III), Mn(IV), Mn(VI), and Mn(VII) [2]. Among the oxidized forms of Mn, divalent Mn(II) is the most soluble species in soils and is also the most available form of Mn for plant acquisition. The solubility of Mn is strongly influenced by soil pH and redox conditions [1,3]. At neutral or higher soil pH, Mn(III) and Mn(IV) are the predominant and insoluble forms of Mn. However, in poorly drained acid soils with pH levels below 5.0 and a reducing environment, oxidized Mn is easily reduced to divalent Mn [4]. Thus, the available Mn in soils is variable and generally ranges from 450 to 4000 mg per kilogram [3]. For example, the concentration of Mn varies between 40 and 1681 mg per kilogram in farmland soils across mainland China [5], while the concentration of Mn in the agricultural soils of central Greece is from 685 to 1307 mg per kilogram [6].

Mn is an example of a transition element that is required for humans, animals, and plants. For most plants, Mn is absolutely necessary at low levels of 20–40 mg per kilogram dry weight [7,8]. Mn is involved in a variety of metabolic processes, including photosynthesis, respiration, fatty acid and protein synthesis, as well as enzyme activation. For example, Mn is an indispensable constitutive element in the Mn cluster structure of the oxygen-evolving complex in photosystem II (PSII) that participates in the water-splitting process, providing necessary electrons for photosynthesis [9,10]. Mn acts as an important cofactor of various enzymes, including superoxide dismutase (MnSOD), catalase (MnCAT), decarboxylases of the tricarboxylic acid (TCA) cycle, and RNA polymerases [8,11]. In addition, Mn is required for multiple steps in the biosynthesis of secondary metabolites, such as lignins, flavonoids, cinnamic acid, and acyl lipids [12].

Despite its necessity, Mn is also considered one of the heavy metals that can be harmful to plants at excessive levels. When the Mn concentration in the aboveground tissues of plants reaches 150 mg per kilogram dry weight, Mn toxicity can generally occur, especially for plants growing in acid soils [13,14]. Many previous studies demonstrate that Mn toxicity can disrupt various physiological processes in plant cells, such as triggering oxidative stress, inhibiting enzyme activity, impeding chlorophyll biosynthesis and photosynthesis, and preventing the uptake and translocation of other mineral elements, including phosphorus (P), Fe, and magnesium (Mg) [14,15,16]. As a result, Mn toxicity leads to the appearance of toxicity symptoms, including chlorosis in young leaves, necrotic dark spots on mature leaves, and crinkled leaves, ultimately inhibiting plant growth. Symptoms of Mn toxicity vary widely among plant species and varieties. For example, chlorosis and necrosis have been reported in leaves of common bean (*Phaseolus vulgaris*) [17], clover (*Trifolium repens*) [18], ryegrass (*Lolium perenne*) [19], and stylo (*Stylosanthes guianensis*) [20]. Brown spots surrounded by irregular areas of chlorotic tissues are observed in cowpea (*Vigna unguiculata*) [21], soybean (*Glycine max*) [22], and barley (*Hordeum vulgare*) [23]. The diverse expressions of Mn toxicity probably indicate different Mn-tolerant capabilities among plant species and cultivars. For example, among different legumes, *Medicago sativa*, *Trifolium fragifevum*, *Leucaena leucocephala*, and *Medicago tvuncatula* are considered the most sensitive to Mn toxicity, while *Centrosemapubescens*, *Lotononis bainesii*, Townsville stylo (*Stylosanthes humilis*), and *Desmodium mcinatum* are the most tolerant plant species [24].

Over the last few decades, there have been major advances in elucidating the mechanisms underlying plant tolerance to Mn toxicity at multiple levels, from physiological changes to biochemical responses (Figure 1). For example, activation of the antioxidant system, including the free radical-mitigating antioxidant enzymes and nonenzymatic components, is thought to be vital for plants alleviating excess Mn-induced oxidative stress [25]. The important roles of the regulation of Mn uptake, translocation, and distribution have been implicated in many plants’ responses to Mn toxicity, such as rice (*Oryza sativa*) [26,27], Arabidopsis (*Arabidopsis thaliana*) [28], and Caribbean stylo (*Stylosanthes hamata*) [29]. Furthermore, plants can sequester Mn into subcellular compartments, such as vacuoles, the endoplasmic reticulum (ER), Golgi apparatuses, and cell walls, to withstand the toxic effects of high Mn stress [30,31]. In addition, free Mn ions can be chelated with protein-based, organic, and inorganic compounds to form inactive Mn complexes, combating the deleterious effects of Mn toxicity [18,19,20].

To date, a variety of genes and proteins have been shown to be involved in the responses to Mn toxicity of plants, such as orange (*Citrus sinensis*) [32], common bean [33], tomato (*Solanum lycopersicum*) [34], stylo [20,35], cowpea [21,36], soybean [22], rice, and barley [23]. Many of the identified genes have been functionally integrated into specific pathways, illuminating the molecular processes of the plant response to Mn toxicity. Furthermore, the functions of numerous genes involved in Mn detoxification through regulation of Mn uptake, distribution, and accumulation have been well characterized in plants [29,37,38,39]. Therefore, the purpose of this review is mainly to focus on Mn as a toxic transition metal to plants and the mechanisms of plant tolerance to Mn stress. This review will discuss the current understanding of plant genes involved in Mn uptake, distribution, and accumulation, which contribute to Mn detoxification. Furthermore, we also highlight the candidate genes that can potentially be used for breeding crop varieties tolerant to Mn toxicity via genetic improvement approaches.

## 2. Activation of the Antioxidant System

As a toxic metal, excess Mn can generate reactive oxygen species (ROS) and trigger oxidative stress in plants, causing lipid peroxidation and damaging photosynthetic pigments and proteins if ROS are not well scavenged [25,35]. One of the adaptive changes that alleviates the toxic effects of high Mn in plants involves the activation of the antioxidant system via antioxidant enzymes, such as superoxide dismutase (SOD), peroxidase (POD), catalase (CAT), ascorbate peroxidase (APX), and glutathione reductase (GR), and nonenzymatic antioxidant components, including ascorbate (AsA) and glutathione (GSH) [35,40]. Increases in the activities of antioxidant enzymes under Mn toxicity are generally associated with enhanced Mn tolerance in common bean [41], cucumber (*Cucumis sativus*) [42], and perennial ryegrass [40]. In perennial ryegrass, for example, the Mn-tolerant ryegrass cultivar Kingston exhibits higher SOD activity than the Mn-sensitive ryegrass cultivar Nui—a higher expression of the *Fe–SOD* gene is observed in Kingston compared to that in Nui [40]. Thus, the induced *Fe–SOD* expression in Kingston is likely to contribute to its high Mn-toxicity tolerance. Additional studies in cowpea have shown that both the activities of H_2_O_2_-producing and H_2_O_2_-consuming PODs are enhanced by Mn toxicity in the leaf apoplast [21]. Furthermore, proteomic analysis indicated that the protein accumulation of PODs in the leaf apoplast is increased by high Mn [21]. Similar results have been implicated in citrus and stylo, in which the expression of *POD* genes is enhanced when plants are subjected to Mn toxicity [35,43]. Therefore, it is probable that SOD and POD represent two key proteins in the plant defense against oxidative damage caused by Mn toxicity. However, considering the damage caused by Mn toxicity, ROS-scavenging systems, through regulation of the antioxidant system, seem to be insufficient to alleviate oxidative stress, which might be a general response of plants to Mn toxicity.

## 3. Regulation of Mn Uptake

Although Mn is required in relatively small amounts, the Mn content accumulated in most plants is approximately 30–500 mg per kilogram dry weight, which is higher than their normal growth requirements [8,14,44]. Therefore, it is reasonable to propose that there are some key transporter genes responsible for Mn acquisition in response to high Mn stress (Figure 2). Studying the mechanisms of plant Mn transport can greatly increase our understanding of how plants acquire and transport Mn under variable environmental Mn levels.

The major transporters responsible for Mn acquisition in plants are members of the natural resistance-associated macrophage protein (Nramp) family, which have so far been functionally characterized in many plants, for example, AtNramp1 from Arabidopsis, OsNramp5 from rice, and HvNramp5 from barley [26,45]. In Arabidopsis, AtNramp1, belonging to the Nramp family, is the major high-affinity Mn transporter involved in Mn uptake. AtNramp1 is localized to the plasma membrane. The transcripts of *AtNramp1* are mainly detected in roots, where their levels are ten times greater than in shoots. Furthermore, *AtNramp1* transcripts are increased by Mn deficiency in the roots [46]. AtNramp1 can complement the phenotype of a yeast mutant, *smf1*, which is defective in Mn uptake when grown in medium containing the divalent cation chelator EGTA [47]. Furthermore, when cultivated in a medium lacking Mn, the T-DNA insertion mutant Atnramp1-1 produces less biomass than wild-type Arabidopsis. The growth inhibition of the mutant can be attributed to less Mn accumulation compared to the wild-type plants under Mn-deficient conditions [46].

In rice, Mn uptake is mediated by OsNramp5, a homolog of AtNramp1 [48]. In contrast to Arabidopsis, OsNramp5 is constitutively expressed in roots, and its expression is enhanced by Fe and zinc (Zn) deficiency but does not respond to different Mn levels in roots [48]. As OsNramp5 can complement the growth of yeast strains defective in Mn and Fe transport, OsNramp5 is implicated in Mn and Fe transport [49]. As OsNramp5 is polarly located at the distal side of both the exodermis and endodermis of mature roots, OsNramp5 is likely to act as an influx transporter and acquire Mn from the soil to the exodermal cells as well as from the apoplastic solution to endodermal cells [48]. Knockout of *OsNramp5* resulted in a decreased concentration of Mn and Fe but not Zn in the shoots, suggesting that OsNramp5 is able to transport Fe in addition to Mn. However, the growth of *OsNramp5* knockout lines is unaffected when the Fe concentration in the external solution is decreased, and the Fe concentrations in the shoots and roots are similar to those of the wild type under Fe deficiency. Thus, the authors conclude that the uptake of Fe required for growth is mediated by other transporters, and OsNramp5 is responsible for additional Fe uptake [48]. A similar key role has been assigned to metal tolerance protein 9 (OsMTP9), the other type of transporter belonging to the cation diffusion facilitator (CDF) family that participates in Mn uptake and translocation in rice roots [50]. *OsMTP9* shows higher expression in roots, but its expression is not influenced by external Mn levels [50]. Tissue- and cell-specific localization analysis revealed that OsMTP9 is localized to the proximal sides of both the exodermis and endodermis of mature root zones, which is opposite to the sites of of OsNramp5 localization in rice roots. Further evidence shows that OsMTP9 acts as an efflux transporter and is responsible for Mn translocation to the root stele [50]. Therefore, the different polar localizations of OsNramp5 and OsMTP9 facilitate Mn uptake from the soil solution to the stele in rice.

Similar results have also been found for HvNramp5, which is localized to the plasma membranes of the epidermal cells of the root tips in the outer root cell layers of barley [51]. There is evidence that HvNramp5 displays transport activity for both Mn and cadmium (Cd) when expressed in yeast cells, and disruption of *HvNramp5* results in growth reduction in barley under low Mn supply [51]. Therefore, HvNramp5 is a transporter required for Mn uptake in barley. In addition, GmDMT1 (divalent metal transporter 1), a nodule-enhanced transporter belonging to the Nramp family in soybean, has also been found to transport Mn in addition to Fe when expressed in yeast [52], although further investigation is needed to understand the physiological roles of GmDMT1 in Mn acquisition in soybean. In addition, members of the zinc-regulated transporter/iron-regulated transporter-like proteins (ZRT/IRT) family were found to have the ability to transport Mn, such as HvIRT1 from barley [53].

Considering the particular importance of the transporter genes controlling Mn uptake in plants, it is reasonable to propose that increased Mn detoxification can be achieved through decreased Mn accumulation from decreasing excess Mn uptake and root-to-shoot Mn translocation, by downregulating transporter genes specific for Mn uptake under high Mn stress. Therefore, manipulation of these transporter genes is an alternative strategy to facilitate the plant response to varying Mn levels through regulation of Mn acquisition.

## 4. Regulation of Mn Translocation and Distribution

After Mn is taken up by roots, most Mn is translocated from roots to shoots and further delivered to various tissues for growth requirements. Thus, it is important to understand the long-distance and whole-plant translocation of Mn in plants in response to different Mn levels, from limited to excessive. In Arabidopsis, two ZIP members, AtZIP1 and AtZIP2, are implicated in Mn translocation from roots to shoots [54]. Both AtZIP1 and AtZIP2 are mainly expressed in the root stele and do not respond to external Mn levels at the transcriptional level. AtZIP1 and AtZIP2 localize to the tonoplast and plasma membrane, respectively. It is probable that AtZIP1 functions in the remobilization of Mn from vacuoles to the cytoplasm in root stellar cells, while AtZIP2 plays a role in Mn movement to the root vasculature for further translocation to the shoots [54]. The loss-of-function mutants of the *AtZIP1* gene in Arabidopsis show severe sensitivity to Mn deficiency. However, the T-DNA *AtZIP2* knockout lines display more tolerance to Mn toxicity than the wild type [54]. Furthermore, Mn concentration in the roots of *AtZIP2* knockout lines is much higher than that in wild-type plants, but no significant differences in shoot Mn concentrations are observed between knockout lines and wild-type plants [54]. Considering that AtZIP2 has high root expression in the stele, AtZIP2 is likely to play a role in Mn transport into the root vasculature, which ultimately helps to provide Mn to the xylem parenchyma, where other transporters such as the heavy metal ATPase, AtHMA2/4, may mediate xylem loading of Mn to the shoot in the transpiration stream as proposed by the authors [54].

OsYSL2, belonging to the yellow stripe-like family, has been characterized to function in long-distance Mn transport and distribution in rice [55]. *OsYSL2* is mainly expressed in leaves, flowers, and developing seeds [55,56]. Electrophysiological measurements using *Xenopus laevis* oocytes show that OsYSL2 is involved in the transportation of Mn–nicotianamine (NA) in addition to Fe–NA complexes [56]. The phloem and seed localization of OsYSL2 suggests that OsYSL2 transports Mn–NA and Fe–NA complexes via the phloem and then loads these complexes into the grain [56]. Overexpression of *OsYSL2* leads to increases in Mn accumulation in the grain [55], suggesting that OsYSL2 is involved in the translocation of Mn into the grain. In addition, some evidence suggests that Mn complexes may be delivered by other transporters, such as *AtOPT3* (a putative oligopeptide transporter) and *AtYSLs* from Arabidopsis [57,58,59], and *ZmYS1* from maize [60], but the exact roles of these genes remain to be clarified.

Additional studies have shown that rice OsNramp3 is a plasma membrane-localized influx transporter for the distribution of Mn, but not Fe and Cd [37]. *OsNramp3* displays higher expression in the nodes and is not affected by external Mn at the transcriptional level. It is noteworthy that the OsNramp3 protein is rapidly degraded within a few hours when plants are exposed to high Mn stress [37]. OsNramp3 is proposed to function with the following patterns: Under Mn deficiency, OsNramp3 preferentially transports Mn to young leaves and panicles via intervascular transfer, but in contrast, under excess Mn conditions, due to rapid OsNramp3 protein degradation, Mn is delivered to old tissues, protecting developing tissues from the toxic effects of excess Mn [37]. Therefore, the authors suggest that OsNramp3 functions as a node-based switch for Mn distribution, which turns the protein on or off in response to variable environmental Mn levels. These findings above provide a major advancement in the understanding of Mn distribution in plants through the regulation of transporters at the post-translational level.

## 5. Intracellular Mn Detoxification in Subcellular Compartments

As the amount of Mn accumulated in most plants usually exceeds their normal growth requirements, plants must cope with excess Mn via internal detoxification. In this regard, one of the key strategies for plant tolerance to Mn toxicity is the compartmentalization of Mn into subcellular compartments [14]. Therefore, transporters that localize to the endomembrane compartments are suggested to be critical for intracellular Mn detoxification in plant cells.

The vacuole, an organelle that comprises approximately 90% of the total cell volume, is the dominant sink for various toxic compounds, including Mn [61]. Some transporters belonging to the CDF family act as proton antiporters for efflux metals (e.g., Zn, Fe, Mn, and Cd) out of the cytoplasm or into subcellular compartments (e.g., vacuoles) [62]. ShMTP1, the first functionally characterized CDF for Mn transport into the vacuoles, was isolated from Caribbean stylo, a tropical legume with superior Mn tolerance [29,35,45]. Evidence shows that ShMTP1 is localized to the tonoplast, and overexpression of *ShMTP1* confers Mn tolerance in yeast cells and Arabidopsis via sequestration of Mn into the vacuoles [29]. In addition to ShMTP1, other CDF members, such as OsMTP8.1, also participate in delivering Mn to vacuoles for Mn sequestration [63]. The transcript of *OsMTP8.1* is mainly detected in shoots and is enhanced by high Mn levels. OsMTP8.1 is expressed in all cells of leaf blades and is also localized to the tonoplast. In rice, knockout of *OsMTP8.1* results in the generation of symptoms of Mn toxicity when plants are exposed to high Mn toxicity [63]. However, OsMTP8.1 is not a unique CDF in mediating Mn transport into vacuoles in rice. OsMTP8.2, a homolog of OsMTP8.1, is also involved in Mn sequestration, and loss of function of *OsMTP8.2* results in severe growth inhibition of both shoots and roots of the *osmtp8.1* mutant in the presence of high Mn [64]. Therefore, it is probable that OsMTP8.2 mediates Mn tolerance together with OsMTP8.1 by sequestering Mn into vacuoles. To date, a set of *MTP* homolog genes have been characterized with similar functions in sequestering Mn into vacuoles, such as *AtMTP8* from Arabidopsis [37], *CsMTP8*/*9* from cucumber [65,66], and *CsMTP8* from the tea plant (*Camellia sinensis*) [67]. The conserved function of MTPs among different plant species fully supports the dominant roles of MTPs in Mn detoxification.

Another major transporter for intracellular Mn sequestration into vacuoles is a member of the cation exchanger (CAX) family with metal/H^+^ antiport activity. In Arabidopsis, the role of AtCAX2 in Mn transport was confirmed by its ability to confer tolerance to Mn toxicity when its expression was heterologous in *pmc1vcx1cnb*, a Mn-sensitive yeast mutant. A three-amino acid Mn-binding region (Cys–Ala–Phe) in AtCAX2 was subsequently found to be critical for Mn-transport activity [68,69,70]. Further analysis showed that overexpression of *AtCAX2* in tobacco (*Nicotiana tabacum*) increases the resistance to Mn toxicity via mediating the sequestration of Mn into the vacuoles [68]. In addition to AtCAX2, AtCAX4 and AtCAX5, which localize to the vacuolar membrane, also display Mn^2+^/H^+^ antiport activity [71,72]. The transcripts of both *AtCAX4* and *AtCAX5* in roots are increased under conditions of high Mn [71,72,73]. Phenotypic analysis shows that transgenic tobacco overexpressing *AtCAX4* displays tolerance to Mn toxicity, while *AtCAX5* can rescue the growth of Mn-sensitive yeast, suggesting their roles in conferring Mn tolerance [72,74]. Arabidopsis mutants, including *cax1*, *cax2*, *cax3*, *cax1*/*cax2,* and *cax2*/*cax3*, have been generated and analyzed for their growth performances under excess Mn levels. Among these mutants, *cax2* and *cax2*/*cax3* displayed severe sensitivity to high Mn stress [75].

An alternate mechanism of intracellular-Mn tolerance in plants is the sequestration of Mn into the Golgi apparatus or endoplasmic reticulum (ER) [15]. AtMTP11 is suggested to be involved in this process in Arabidopsis. AtMTP11 can rescue the growth of yeast mutant *pmr1*, which is defective in a Ca^2+^/Mn^2+^–ATPase, in the presence of excess Mn. Arabidopsis mutants impaired in *AtMTP11* are sensitive to high Mn levels, whereas plants overexpressing *AtMTP11* are more tolerant to Mn toxicity [76]. In contrast to ShMTP1, OsMTP8.1, and OsMTP8.2 mentioned above, AtMTP11 is localized to a punctate endomembrane compartment probably in the trans-Golgi, but not to the plasma membrane and vacuole. Therefore, a secretory pathway involving vesicular trafficking and exocytosis mediated by AtMTP11 is believed to help increase Mn tolerance in Arabidopsis [28]. Similar functions of other MTPs in sequestering Mn into the Golgi apparatus have been reported for OsMTP11 from rice [27], HvMTP8.1 and HvMTP8.2 from barley [77], PtMTP11.1 and PtMTP11.2 from poplar (*Populus trichocarpa*) [28], as well as BmMTP10 and BmMTP11 from beets (*Beta vulgaris*) [78].

It has been well demonstrated that ER-type calcium ATPases (ECAs), belonging to the Ca^2+^–ATPase subfamily, can use energy from ATP hydrolysis to catalyze the translocation of cations across membranes [79,80]. There are four predicted ECAs in Arabidopsis (AtECA1–4) and three in rice (OsECA1–3) [79]. In Arabidopsis, AtECA1 and AtECA3 are localized to the ER and Golgi compartments, respectively [81,82,83]. The expression of AtECA1 and AtECA3 was found in all major organs of Arabidopsis, especially in the roots [81,83]. Both AtECA1 and AtECA3 are able to rescue the growth of yeast under high Mn stress [81,84]. Furthermore, under excess-Mn conditions, the Arabidopsis *ateca1-1* mutants display inhibited root growth, and the growth of the *ateca1-1* mutant is rescued by overexpression of *AtECA1* [81]. Similarly, the root growth of the *ateca3* mutant is impaired by excess Mn, confirming that AtECA3 is also necessary for Mn detoxification in Arabidopsis [84]. Therefore, AtECA1 and AtECA3 are the two key components required for delivering Mn into the ER and Golgi compartments for Mn tolerance. In addition, the YSL family is also implicated in the sequestration of Mn into endomembrane compartments. AtYSL4 and AtYSL6 are reported to be localized to vacuole membranes and internal membranes resembling the ER in Arabidopsis. Significant decreases in fresh weight have been observed in single mutants and double mutants of *AtYSL4* and *AtYSL6* compared to wild-type Arabidopsis grown in high Mn for 21 d [59]. The authors suggest a role for AtYSL4 and AtYSL6 in the sequestration or efflux of this metal into intracellular compartments [59]. However, future characterization of YSL as well as ECAs in other crop species is needed to confirm their exact roles in Mn detoxification via sequestration of Mn into intracellular compartments.

OsYSL6 is reported to transport Mn from the apoplast to the symplast, which is required for the detoxification of excess Mn in rice [84]. Although the expression of *OsYSL6* does not respond to either deficiency or toxicity of Mn, ectopic expression of *OsYSL6* in the yeast mutant indicates transport activity for the Mn–NA complex. Furthermore, knockout of *OsYSL6* in rice increases Mn accumulation in the leaf apoplast but not in the symplast under high Mn stress, resulting in the development of necrosis in the old leaves, a symptom of Mn toxicity [84]. As divalent Mn accumulated in the apoplast can potentially be oxidized to trivalent Mn, which further oxidizes proteins and lipids, causing deleterious effects of Mn toxicity [21], OsYSL6 is likely to alleviate excess Mn toxicity via the transport of Mn from the apoplast to the symplast in rice. 

Most of the Mn transporter genes mentioned above display no or only slight responses to varying Mn levels, which may partially explain why plants accumulate large amounts of Mn that far exceed their growth requirements. Therefore, it is of great importance to investigate the regulatory mechanisms of the plant response to external Mn in the future.

## 6. Si Application Alleviates Mn Toxicity

Another strategy for increasing Mn tolerance can be achieved by the application of silicon (Si) to the roots of plants such as rice [85], cowpea [86,87], and cucumber [88]. The mechanisms for Si-alleviated Mn toxicity include decreasing the Mn accumulation in shoots, promoting Mn oxidation in roots and increasing the cell wall-binding capacity for Mn [88,89,90]. A recent study showed that supplementation with Si successfully decreases the Mn concentration in the shoots but increases Mn in the roots of rice under high Mn stress, alleviating Mn toxicity [90]. However, Si application cannot alleviate Mn toxicity in the rice *lsi1* mutant, which is defective in Si uptake. OsLsi1 is a Si transporter that transports Si from the external solution to the root cells in rice [91]. Interestingly, the expression of *OsNramp5* is decreased by long-term exposure to Si in the wild type but not in the *lsi1* mutant. The authors suggest that the Si-alleviated Mn toxicity in rice can be attributed to inhibition of root-to-shoot translocation of Mn and decreased Mn uptake by downregulation of Mn transporters, such as OsNramp5 and OsMTP9 [90]. Therefore, OsLsi1 might participate in Mn detoxification through regulation of Si uptake, which deserves further clarification.

## 7. Organic Acid Mediates Mn Detoxification

Mn can be chelated with protein-based, organic, and inorganic compounds to form Mn complexes, thus decreasing Mn uptake and/or Mn phytotoxicity. Regulation of organic acid metabolism is an important strategy in Mn detoxification. Intracellular Mn in cowpea, *Gossia bidwillii*, and *Phytolacca acinosa* is found to be chelated in complexes with internal citrate, malate, and oxalate, respectively [92,93,94]. The complexation of Mn by organic acids in the apoplast is proposed to decrease Mn phytotoxicity in cowpea [87]. Increases in internal malate concentrations are observed in leaves and roots of the Mn-tolerant stylo genotype Fine-stem under high Mn stress, and are closely linked to its Mn tolerance capabilities [20]. Accordingly, Mn might be chelated by malate to form Mn–malate complexes, ultimately conferring Mn tolerance in stylo. Subsequent analysis shows that malate synthesis in stylo could be attributed to a Mn-enhanced malate dehydrogenase (SgMDH1), which catalyzes the reversible conversion of oxaloacetate to malate. Due to successful increases in resistance to Mn toxicity in both yeast cells and Arabidopsis, SgMDH1 is hypothesized to be involved in Mn detoxification through mediated malate synthesis [20].

On the other hand, increases in organic acid exudation from roots in response to Mn toxicity are found in stylo, clover, and ryegrass [18,19,20]. Increased root exudates of oxalate and citrate in Mn-tolerant ryegrass cultivars have been implicated in increasing Mn tolerance by decreasing Mn uptake from the rhizosphere [19]. Similar results are also reported in stylo, where increased malate exudation from roots helps to confer Mn tolerance, and exogenous malate application to the growth medium increases the resistance of the Mn-sensitive stylo genotype to the toxic effects of Mn [20]. Interestingly, the expression of an aluminum-activated malate transporter (*SgALMT1*) is enhanced by high Mn stress in the Mn-tolerant stylo genotype [20], which likely functions in mediating malate efflux from roots, as observed in aluminum detoxification [95]. Therefore, it is reasonable to hypothesize that coordinated regulation of malate synthesis and exudation by *SgMDH1* combined with *SgALMT1* might facilitate the tolerance of stylo to Mn toxicity.

## 8. Other Aspects

In recent years, the development of biotechnologies, such as RNA-seq and proteomics, has provided favorable platforms to reveal complex responses of plants to biotic and abiotic stresses [35,96,97]. Many differentially expressed genes and proteins have been previously identified in plants’ responses to Mn toxicity. For example, various Mn-responsive genes have been isolated from leaves of citrus using cDNA–AFLP technology, and the identified genes can be classified into different functional categories, such as biological regulation and signal transduction (e.g., protein phosphatase 2a and Myb family transcription factor), carbohydrate and energy metabolism (e.g., ATP synthase subunit alpha and UDP-glycosyltransferases), nucleic acid metabolism (e.g., DNA polymerase phi subunit and histone H4), protein metabolism (e.g., ribosomal proteins, eukaryotic initiation factors, and glutathione S-transferase Tau2), cell wall metabolism (e.g., cell wall-associated hydrolase and glycoside hydrolase family 28 protein), stress responses (e.g., *CAT*, *POD42*, and monodehydroascorbate reductase), and cell transport (e.g., ABC transporter family protein) [32]. In addition, a set of Mn-regulated proteins were identified in the Mn-tolerant stylo genotype through proteomic analysis. These proteins are mainly involved in defense responses, photosynthesis, carbon fixation, metabolism, cell wall modulation, and signaling [35]. Further analysis shows that some of the identified proteins related to the phenylpropanoid pathway, including phenylalanine ammonia-lyase (PAL), chalcone synthase (CHS), chalcone–flavonone isomerase family protein (CFI), and isoflavone reductase (IFR), are regulated by external Mn in stylo [35]. As secondary metabolites, such as phenolics, flavonoids, phenylalanine, and callose, have been reported to be regulated by excess Mn in plants [12,16,98], the regulation of the phenylpropanoid pathway seems to facilitate plants’ adaptations to Mn toxicity. Furthermore, combined with the physiological and proteomic analysis, the molecular responses involved in stylo adaptation to Mn toxicity are suggested to include enhancing defense responses and phenylpropanoid pathways, adjusting photosynthesis and metabolic processes, and modulating protein synthesis and turnover [35]. Despite the advances in the identification of various genes and proteins responding to Mn toxicity, there remains a scarcity of work designed to investigate how these genes are involved in plant tolerance to Mn toxicity, and future work is needed in these areas. 

## 9. Future Perspectives

Of the mineral nutrients essential for plant growth, Mn can cause phytotoxicity at excess levels, especially in acid soils. Even with the examination of the physiological and molecular mechanisms and characterization of genes controlling Mn tolerance over the last few decades, relatively little is known about the molecular mechanisms regulating Mn homeostasis and detoxification in plants, which are critical to allow plants to adjust their Mn requirements and to avoid toxicity. Furthermore, as many genes responsible for Mn transport and distribution are not or are only slightly responsive to external Mn, future work is required to elucidate the possible regulatory mechanisms, such as transcriptional regulatory networks and post-translational protein modifications (e.g., phosphorylation, ubiquitination, and glycosylation), by which these components facilitate plant adaptions to changing Mn levels.

Although some genes have been implicated in Mn detoxification via ectopic expression in model yeast cells or Arabidopsis, the exact roles of these genes need to be determined at both the cellular and whole-plant levels, considering molecular and physiological aspects in planta. Aside from the model plants Arabidopsis and rice, candidate genes in other crop species should be identified to clarify their roles in Mn acquisition and detoxification, which might be more complicated depending on the physiological, biochemical, and molecular responses in different crops. Once identified, these genes can potentially be used to breed crop varieties with high Mn acquisition efficiency under Mn deficiency in alkaline soils, or with increased Mn tolerance under Mn toxicity in acid soils. Additionally, in some hyperaccumulator plants that can store high levels of toxic metals without displaying obvious toxicity, excess Mn has been shown to accumulate in the non-photosynthetic tissues for detoxification [99,100]. However, the mechanisms underlying Mn hyperaccumulation and the responses of hyperaccumulators to Mn remain poorly understood. Candidate genes responsible for Mn detoxification in Mn-hyperaccumulator plants have yet to be reported. These are some of the future directions that should be taken into account, as these resources can be exploited to develop genetically engineered plants used for Mn phytoremediation.

To date, most of the studies conducted to investigate gene functions in Mn detoxification have mainly focused on Mn transport, distribution, or homeostasis. Genes associated with other pathways, such as biological regulation and signal transduction, photosynthesis, carbohydrate and energy metabolism, and secondary metabolism, which can potentially influence Mn tolerance mechanisms, have received little attention. Future efforts to investigate these areas are of great importance for increasing our understanding of how plants detoxify Mn.

## 10. Conclusions

Although Mn is an essential element for plants, excess Mn can cause phytotoxicity, inhibiting plant growth. This review shows that increasing plant Mn tolerance can be achieved by coordination of Mn absorption, translocation, and distribution, as well as by complex regulations of physiological changes and biochemical responses. This review highlights that Mn detoxification is regulated by a variety of genes and proteins associated with specific pathways, such as Mn transport and homeostasis, which can potentially be used to breed crop varieties with high Mn tolerance. This review also provides some of the future areas that could be taken into account in terms of gaining a better understanding of how plants tolerate Mn toxicity.

## Figures and Tables

**Figure 1 ijms-20-05096-f001:**
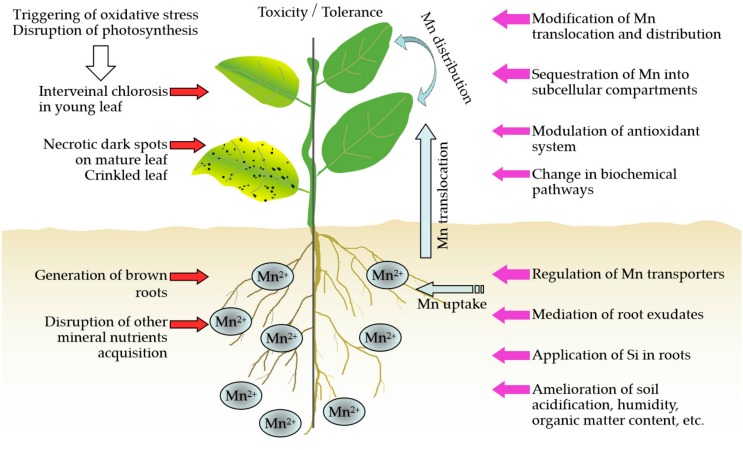
Schematic representation of Mn toxicity and strategies for increasing Mn tolerance in plants. Mn toxicity can trigger oxidative stress and disrupt photosynthesis, which may result in the generation of interveinal chlorosis in young leaves, necrotic dark spots on mature leaves, and crinkled leaf. Furthermore, Mn toxicity can lead to the formation of brown roots and prevent the uptake and translocation of other mineral elements. In plants, Mn tolerance strategies include modification of Mn translocation and distribution, sequestration of Mn into subcellular compartments, modulation of the antioxidant system, changes in biochemical pathways, and regulation of Mn transporters. In addition, the mediation of root exudates, the application of Si in roots, and the amelioration of soil acidification, humidity, and organic matter content also contribute to increase plant Mn tolerance. Red arrows indicate the toxic effects of excess Mn to plants. Purple arrows represent the adaptive strategies of plants to Mn toxicity.

**Figure 2 ijms-20-05096-f002:**
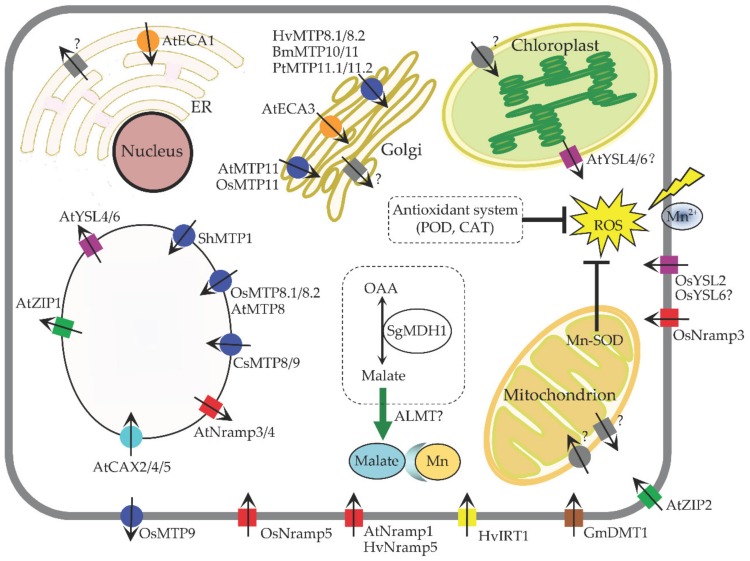
Summary of genes affecting Mn transport and tolerance in plants. Squares: Import into the cytosol; circles: Export out of the cytosol; blue: MTP family; green: ZTP family; red: Nramp family; purple: YSL family; yellow: IRT family; orange: ECA family; cyan: CAX family; brown: DMT family; gray: unknown. ER: Endoplasmic reticulum; Nramp: Natural resistance-associated macrophage protein; MTP: Metal tolerance protein; DMT: Divalent metal transporter; ZIP/IRT: Zinc-regulated transporter/iron-regulated transporter-like proteins; YSL: Yellow stripe-like protein; CAX: Cation exchanger; ECAs: ER-type calcium ATPases; MDH: Malate dehydrogenase; ALMT: Aluminum-activated malate transporter; OAA: Oxaloacetate; ROS: Reactive oxygen species; SOD: Superoxide dismutase; POD: Peroxidase; CAT: Catalase. At: *Arabidopsis thaliana*; Os: *Oryza sativa*; Gm: *Glycine max*; Hv: *Hordeum vulgare*; Mt: *Medicago tvuncatula*; Cs: *Cucumis sativus*; Sh: *Stylosanthes hamata*; Sg: *Stylosanthes guianensis*; Le: *Lycopersicon esculentum*; Bm: *Beta vulgaris* subspecies maritima; Pt: *Populus trichocarpa*. Question marks behind some genes mean that the exact roles of these genes or their localizations remain to be further clarified.

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
