# Peer review of "Advances in the Mechanisms of Plant Tolerance to Manganese Toxicity"

_ijms, 2019, doi:10.3390/ijms20205096_

Round 1

Reviewer 1 Report

Dear Authors,

I had great privilege to assess a review paper entitled “Advances in the mechanisms of plants tolerance to manganese toxicity” (manuscript id: ijms 611523) which is considered for publication in International Journal of Molecular Sciences. This review paper is well written and presents new insights in interesting problem of manganese toxicity. Authors presents with use of great (beautiful) schemes connection of Mn-plant relation. Because of form of presentation use of scientific language and numerous interesting information about Mn I would like to recommend publication in current form.

Best regards,

Author Response

Answer: Thank you for your valuable comments.

Reviewer 2 Report

The manuscript written by Li et al addressed an important topic on the mechanisms of plant tolerance to Manganese (Mn) toxicity. Their review provides certain understanding regarding the adaptive strategies of plants to Mn toxicity and will be particular useful for breeding crop varieties with improved Mn tolerance.

A general comment to the authors would be how you see the difference between this manuscript vs. the papers recently published by Socha and Guerinot (2014), and Fernando and Lynch (2015) that leads to the necessity of finishing this manuscript.

Some specifics regarding the manuscript:

Line 64 to 65: It would be better to add a transition paragraph here to introduce the sensitivity vs. tolerance of the plants to excessive Mn, which will lead to the next paragraph about the researches on mechanisms of tolerance to Mn toxicity.

Line 90 to 106 and line 129 to 144: The figure with its caption should be able to stand on its own. Both figures in the manuscript lack detailed explanations and make little/no sense to the readers if without looking through the text. An explanation of the meanings of different symbols, abbreviations would help the readers to understand.

Figure 1: For example, what the purple arrows and the red arrows mean?

Figure 2: For example, what the symbols like square and circle mean, as well as the different colors of the squares and circles? What the question marks behind some genes mean?

Besides, each gene on Figure 2 should have its full name listed in the figure caption along with the reference it came from. It is very important to acknowledge the work done by other researchers.

Line 438 to 442: The conclusion part was overly simplified. It should highlight what the readers could take back from this manuscript instead of barely any useful information.

Author Response

The manuscript written by Li et al addressed an important topic on the mechanisms of plant tolerance to Manganese (Mn) toxicity. Their review provides certain understanding regarding the adaptive strategies of plants to Mn toxicity and will be particular useful for breeding crop varieties with improved Mn tolerance.

A general comment to the authors would be how you see the difference between this manuscript vs. the papers recently published by Socha and Guerinot (2014), and Fernando and Lynch (2015) that leads to the necessity of finishing this manuscript.

Answer: Thank you for your valuable comments. The paper published by Socha and Guerinot (2014) is mainly to focus on transporter gene families that have been implicated in Mn transport regarding the Mn uptake and mobilization, especially for plants response to Mn deficiency, while the paper published by Fernando and Lynch (2015) is mainly to discuss plant Mn phytotoxicity mechanisms at the physiological levels. More recently, there are other progresses in the mechanisms of plants adaption to Mn toxicity, particular for the dissection of how plants compartmentalization of Mn. Thus, this manuscript summarizes the recent advances and discusses the roles of genes responsible for Mn detoxification. We believe that this review will increase the understanding of how plants adapt to Mn toxicity at some aspects.

Some specifics regarding the manuscript:

Line 64 to 65: It would be better to add a transition paragraph here to introduce the sensitivity vs. tolerance of the plants to excessive Mn, which will lead to the next paragraph about the researches on mechanisms of tolerance to Mn toxicity.

Answer: Thank you for your helpful suggestions. We have added more descriptions about this accordingly. Please check this in details in the revised manuscript.

Line 90 to 106 and line 129 to 144: The figure with its caption should be able to stand on its own. Both figures in the manuscript lack detailed explanations and make little/no sense to the readers if without looking through the text. An explanation of the meanings of different symbols, abbreviations would help the readers to understand.

Answer: Thank you for your helpful suggestions. You are right. We have added figure legends accordingly. Please see the revised manuscript.

Figure 1: For example, what the purple arrows and the red arrows mean?

Answer: Thank you for your helpful suggestions. Red arrows on the left side indicate the toxic effects of excess Mn to plants, while purple arrows on the right side represent the adaptive strategies of plants to Mn toxicity. We have revised the manuscript accordingly. Please see the revised manuscript.

Figure 2: For example, what the symbols like square and circle mean, as well as the different colors of the squares and circles? What the question marks behind some genes mean?

Answer: Thank you for your helpful suggestions. Squares, import into the cytosol. Circles, export out of the cytosol. Question marks behind some genes indicate the exact roles of these genes or its localization remain to be further clarified. We have revised the manuscript accordingly. Please see the revised manuscript.

Besides, each gene on Figure 2 should have its full name listed in the figure caption along with the reference it came from. It is very important to acknowledge the work done by other researchers.

Answer: Thank you for your helpful suggestions. We have revised the figure legends accordingly. Yet, it is better to cite the corresponding reference in the text according to the journal. Please see the revised manuscript.

Line 438 to 442: The conclusion part was overly simplified. It should highlight what the readers could take back from this manuscript instead of barely any useful information.

Answer: Thank you for your valuable comments. We have revised the conclusion accordingly. Please see the revised manuscript.

Reviewer 3 Report

Overall, this review is a good contribution to understanding the mechanisms of plants tolerance to manganese toxicity. Two main issues may be raised: 1) some sections contradict themselves with what has been said previously (see comments below); 2) Authors recognized that several of the genes reported in the review are from studies in yeast, which to this reviewer should be avoided unless there are studies showing their function in plants (preferably beyond arabidopsis). Other comments:

Lines 41-44: Authors should include a range of Mn instead of an average because it is meaningless without presenting the spread of concentrations.

Line 52: spell out TCA.

Line 71: include Latin name for Arabidopsis.

Line 78: remove the ‘,’ (semicolon) after ‘plants’. Also C. sinensis = orange?

Fig. 1: caption should explain more what is being presented. (red arrows or left side indicate…) same with the other side.

Fig. 2: Same comment as for fig. 1. Authors should provide a better caption. In some organelles, there is only one direction for the ‘gene’ are there also transporters involved in extruding Mn for instance from the golgi element?. The cell membrane only has genes regulating uptake of Mn? What is happening with SgMDH1?. In addition, in the mitochondria, is it Mn-SOD or FeSOD? Below in the text it says that OsMTP9 is involved in efflux (but shown as involved in uptake in Fig. 2) and in the proximal side of the plasma membrane. This could be reflected in this figure.

Lines 148-154: this section is repetitive and could be reduced.

Lina 159: is Nramp expressed only in roots or in the entire plant?

Line 162-165: this sentence needs clarity. It seems that it contradicts itself.

Line 168: does OSNramp5 transport Fe and Zn as well? This has to be clear because it is mentioned again in Line 175 (why not Fe then?).

Line 184: found for instead of found that.

Lines 196-201: would downregulation of these genes affect then Zn and Fe uptake? Based on what is described above, it may be an issue. How authors propose to overcome this potential tradeoff?

Lines 208-216:  AtZIP1 shouldn’t be considered as contributing to the transport of Mn to the xylem. The last sentence doesn’t make sense if AtZIP2 mutants become more tolerant under Mn toxicity, it means that Mn is not transported to the shoot, which provides ‘tolerance’. The last sentence talks about sequestration in the shoot, but this has not been discussed here as a role for AtZIP2. Which is the gene that moves Mn into the vacuoles?

Lines 235-237: This sentence contradicts what was said before. OsNramp3 doesn’t transport Mn from young to old tissue. It degrades at high Mn concentrations. What is “adapted to environmental Mn availability”? the latter needs clarity.

Line 238: Which study?

Lines 240-242: delete. This has been said throughout the manuscript and it is the purpose of the review. It doesn’t add anything.

Lines 319-320: How can ‘detoxification’ happen when Mn is ‘sequestered’ in the cytoplasm?

Lines 330: How can authors referred to accumulation of Mn in the apoplast as compartmentalization? Although it can be a ‘compartment’ it is still space outside the plasma membrane where solutes/water moves freely (or with less resistance than in the symplast).

Lines 331-333: It has been argued in this manuscript that Mn needs to be compartmentalized in organelles such as vacuoles. How do the authors suggest that the apoplast can be an area to accumulate Mn to reduce toxicity?

Author Response

Overall, this review is a good contribution to understanding the mechanisms of plants tolerance to manganese toxicity. Two main issues may be raised: 1) some sections contradict themselves with what has been said previously (see comments below); 2) Authors recognized that several of the genes reported in the review are from studies in yeast, which to this reviewer should be avoided unless there are studies showing their function in plants (preferably beyond arabidopsis).

Answer: Thank you for your valuable comments and suggestions. We have addressed your comments, and have deleted the genes from studies in yeast accordingly. Please see the revised manuscript.

Other comments:

Lines 41-44: Authors should include a range of Mn instead of an average because it is meaningless without presenting the spread of concentrations.

Answer: Thank you for your helpful suggestions. We have revised this accordingly. Please see the revised manuscript.

Line 52: spell out TCA.

Answer: Thank you for your suggestions. We have revised this accordingly. Please see the revised manuscript.

Line 71: include Latin name for Arabidopsis.

Answer: Thank you for your suggestions. We have revised this accordingly. Please see the revised manuscript.

Line 78: remove the ‘,’ (semicolon) after ‘plants’. Also C. sinensis = orange?

Answer: Thank you for your suggestions. We have removed the ‘,’ after ‘plants’. Furthermore, Citrus sinensis represents orange. We have revised this accordingly. Please see the revised manuscript.

Fig. 1: caption should explain more what is being presented. (red arrows or left side indicate…) same with the other side.

Answer: Thank you for your helpful suggestions. We have added the figure legends accordingly. Please see the revised manuscript. 

Fig. 2: Same comment as for fig. 1. Authors should provide a better caption. In some organelles, there is only one direction for the ‘gene’ are there also transporters involved in extruding Mn for instance from the golgi element?. The cell membrane only has genes regulating uptake of Mn? What is happening with SgMDH1?. In addition, in the mitochondria, is it Mn-SOD or FeSOD? Below in the text it says that OsMTP9 is involved in efflux (but shown as involved in uptake in Fig. 2) and in the proximal side of the plasma membrane. This could be reflected in this figure.

Answer: Thank you for your helpful suggestions. You are right. There are some transporter genes involved in extruding Mn from the organelles and also from the cell membrane, in despite of without being functional characterized. The Mn tolerance related gene, SgMDH1, has been reported to be enhanced by excss Mn in roots of stylo that catalyzes the reversible conversion of oxaloacetate to malate, which might subsequent chelate Mn to form Mn-malate complexes. Thus, we included it in the figure. In the mitochondria, it is Mn-SOD. In addition, OsMTP9 acts as an efflux transporter. We have revised the figure and added the figure legends accordingly. Please see the revised manuscript.

Lines 148-154: this section is repetitive and could be reduced.

Answer: Thank you for your helpful suggestions. We have revised this section accordingly. Please see the revised manuscript.

Lina 159: is Nramp expressed only in roots or in the entire plant?

Answer: Thank you for your helpful comments. We have added this description accordingly. Please see the revised manuscript.

Line 162-165: this sentence needs clarity. It seems that it contradicts itself.

Answer: Thank you for your valuable comments. We have revised these accordingly. Please see the revised manuscript.

Line 168: does OSNramp5 transport Fe and Zn as well? This has to be clear because it is mentioned again in Line 175 (why not Fe then?).

Answer: Thank you for your valuable comments. The authors found that OsNramp5 expression is enhanced by Fe and Zn deficiency but not respond to different Mn levels in roots. As OsNramp5 can complement the growth of yeast strains defective in Mn and Fe transport, OsNRAMP5 is implicated in Mn and Fe transport. Furthermore, knockout of OsNramp5 resulted in a decreased concentration of Mn and Fe but not Zn in the shoots, suggesting that OsNramp5 is able to transport Fe in addition to Mn. However, the growth of OsNramp5 knockout line is unaffected when decreased Fe concentration in the external solution, and Fe concentration in the shoots and roots is similar to that of the wild type under Fe deficiency. Thus, the authors conclude that the uptake of Fe required for growth is mediated by other transporters and OsNramp5 is responsible for additional Fe uptake. We have revised this accordingly. Please see the revised manuscript.

Line 184: found for instead of found that.

Answer: Thank you for your helpful suggestions. We have revised this accordingly. Please see the revised manuscript.

Lines 196-201: would downregulation of these genes affect then Zn and Fe uptake? Based on what is described above, it may be an issue. How authors propose to overcome this potential tradeoff?

Answer: Thank you for your valuable comments. You are right. Downregulation of Mn transporter genes may affect other mineral nutrients (e.g., Zn and Fe) uptake, such as OsNramp5. Thus, some kinds of transporters specific for Mn uptake can be considered as the candidate genes used for manipulation without affecting other nutrients uptake. We have revised this accordingly. Please see the revised manuscript. 

Lines 208-216: AtZIP1 shouldn’t be considered as contributing to the transport of Mn to the xylem. The last sentence doesn’t make sense if AtZIP2 mutants become more tolerant under Mn toxicity, it means that Mn is not transported to the shoot, which provides ‘tolerance’. The last sentence talks about sequestration in the shoot, but this has not been discussed here as a role for AtZIP2. Which is the gene that moves Mn into the vacuoles?

Answer: Thank you for your valuable comments and suggestions. You are right. We have deleted the sentence ‘contributing to Mn transport to the xylem parenchyma’. In the study of AtZIP2, the T-DNA AtZIP2 knockout lines display more tolerant than wild type to Mn toxicity [56]. Furthermore, Mn concentration in root of the AtZIP2 knockout lines exhibit much higher than that of wild type plants, but no significant differences in shoot Mn concentration are observed between knockout lines and wild type plants [56]. Considering AtZIP2 with high root expression in the stele, AtZIP2 is likely to play a role in Mn transport into the root vasculature, which ultimately helps to provide Mn to the xylem parenchyma, where other transporters such as heavy metal ATPase, AtHMA2/4, mediate xylem loading of Mn to the shoot in the transpiration stream as proposed by the authors [56]. This is the role suggested for AtZIP2 in Mn translocation. In addition, high levels of root Mn can be moved into the vacuoles through the function of some transporters, such as AtMTP8, conferring Mn tolerance. We have revised these accordingly. Please see the revised manuscript.

Lines 235-237: This sentence contradicts what was said before. OsNramp3 doesn’t transport Mn from young to old tissue. It degrades at high Mn concentrations. What is “adapted to environmental Mn availability”? the latter needs clarity.

Answer: Thank you for your valuable comments. You are right. We have revised these accordingly. Please see the revised manuscript.

Line 238: Which study?

Answer: Thank you for your valuable comments. The study is functional characterization of OsNramp3 (Yamaji et al., 2013). We have revised this accordingly. Please see the revised manuscript.

Lines 240-242: delete. This has been said throughout the manuscript and it is the purpose of the review. It doesn’t add anything.

Answer: Thank you for your helpful suggestions. We have deleted this paragraph accordingly. Please see the revised manuscript.

Lines 319-320: How can ‘detoxification’ happen when Mn is ‘sequestered’ in the cytoplasm?

Answer: Thank you for your valuable comments and suggestions. We have revised this sentence accordingly. Please see the revised manuscript.

Lines 330: How can authors referred to accumulation of Mn in the apoplast as compartmentalization? Although it can be a ‘compartment’ it is still space outside the plasma membrane where solutes/water moves freely (or with less resistance than in the symplast).

Answer: Thank you for your valuable comments. We agree with you. Here seems to confuse. We have deleted these descriptions accordingly. Please see the revised manuscript.

Lines 331-333: It has been argued in this manuscript that Mn needs to be compartmentalized in organelles such as vacuoles. How do the authors suggest that the apoplast can be an area to accumulate Mn to reduce toxicity?

Answer: Thank you for your valuable comments. You are right. We have deleted these descriptions accordingly. Please see the revised manuscript.